# A Novel Method for Gait Analysis on Center of Pressure Excursion Based on a Pressure-Sensitive Mat

**DOI:** 10.3390/ijerph17217845

**Published:** 2020-10-26

**Authors:** Eun-tae Jeon, Hwi-young Cho

**Affiliations:** 1Department of Health Science, Gachon University Graduate School, Incheon 21936, Korea; gksmfskdls@gmail.com; 2Department of Neurology, Korea University Ansan Hospital, Korea University College of Medicine, Ansan 15355, Korea; 3Medical Science Research Center, Korea University Ansan Hospital, Korea University College of Medicine, Ansan 15355, Korea; 4Department of Physical Therapy, Gachon University, Incheon 21936, Korea

**Keywords:** gait, pressure, reference values, rehabilitation measurement, reproducibility of results

## Abstract

Center of pressure (COP) during gait is a useful measure for assessing gait ability and has been investigated using platform or insole systems. However, these systems have inherent restrictions in repeated measure design or in obtaining true vertical force. This study proposes a novel method based on a pressure-sensitive mat system for COP measurement and presents normal reference values for the system. To explore repeatability, this work also investigated relative and absolute intra-rater reliabilities and determined the number of footfalls required to obtain a reliable measurement. Ninety healthy young adults participated and performed barefoot walking on a force-sensitive mat at a comfortable and fast pace. The time points and subphase duration of the stance phase, displacement ranges, and mean locations of COP and velocity of COP excursion were parameterized. The results showed acceptable and consistent variabilities of the parameters. Seven footfalls were determined as the threshold for most parameters to show a good to reasonable level of reliability. In conclusion, the presented method can be used as a reliable measurement for COP excursion, and it is recommended that more than seven footfalls be collected to ensure a high level of reliability.

## 1. Introduction

Center of pressure (COP) is defined as the centroid of the ground reaction force vector. COP from the vertical ground reaction force acting on the plantar surface during the stance phase is a common measure in gait analysis. Dynamic properties of COP excursion reveal balance ability [1], gait performance [2], intervention efficacy [3], and foot and ankle function [4].

Previous studies on measurement of COP have employed an in-shoe system and a platform system. The former has the unique advantage of providing information at the foot–shoe interface; however, it measures distorted vertical force, especially during the initial and late stages of the stance phase due to its position [5]. The latter provides accurate force data, as it is embedded in the ground; however, its restricted size often results in subjects having difficulty maintaining contact with the plate. Their feet must be in contact with the plate entirely to obtain complete data; unconscious “targeting” can alter the subject’s natural gait pattern [5,6,7]. Furthermore, excessive repetition may increase the risk of plantar ulceration in patients with diabetes and peripheral neuropathy [5]. A repeated measure design may also be unrealistic for patients with neurological impairments such as proprioception or coordination disorders [5].

The advantages of a pressure-sensitive mat system are evident, as the mat is relatively large and parallel to the ground. The mat enables users to obtain data from multiple footfalls at a time, and it allows various analyses such as investigation of the possible interactive effects between the feet or an individual’s gait variability. One study attempted to quantify COP progression during gait with a mat system in individuals with stroke [8]. The authors investigated the anteroposterior velocities of COP progression in a foot segmented into three equal regions along the length. However, the analysis was performed only on a single limb in the stance phase, and control of mediolateral stability was represented with only a single COP location.

Therefore, the primary purpose of the present study was to propose a COP measurement method based on a pressure-sensitive mat enabling examination of more detailed characteristics of the COP excursion throughout the stance phase. This work also focused on the establishment of normal reference values and the investigation of relative and absolute intra-session reliabilities of the measurement. The secondary purposes of the study were to determine the minimum number of footfalls needed to obtain reliable measures and to investigate normal foot dominance and gait speed effects on variability of COP excursion.

## 2. Methods

### 2.1. Participants

Participants included those candidates between 19 and 29 years old who at the time of the study did not have any of the following: (1) pain or discomfort during gait; (2) history of lower limb injury affecting activities for more than 24 h within the past 6 months; (3) diagnosis of a neurological disorder; and (4) fractures, deformities, or surgeries affecting the lower limbs. This study was approved by the Institutional Review Board of Gachon University and registered with the Korea Clinical Trials Registry (KCT0002519, 11 March 2017). All participants provided informed consent prior to participation, and all methods were performed in accordance with the approved experimental protocols and regulations.

### 2.2. Measures and Procedures

The height of the medial longitudinal arch of the foot was included in the anthropometric measurements and was assessed using the arch index. The arch index is determined by measuring the ratio of the area of the middle one-third from a footprint excluding the toes (Figure 1) [9]. The arch index is highly correlated with navicular height [10] and pressure under the midfoot during walking [11]. A plantar pressure distribution measurement system (FDM-S; Zebris Medical GmbH, Weitnau-Seltmans, Germany) was used to obtain footprints with a sampling frequency of 100 Hz. This system and the selected measurement parameters are known to have good repeatability (intraclass correlation coefficients (ICC) = 0.85–0.91) [12]. MATLAB software (MATLAB R2018A; The MathWorks Inc., Natick, MA, USA) was used for image processing and computation.

GAITRite (CIR Systems Inc., Sparta, NJ, USA), which is a 4.6-m-long force-sensitive mat with an active area 3.66 m × 0.61 m wide and a spatial resolution of 1.27 cm, was used to measure COP excursion with a sampling frequency of 240 Hz. Lines were marked 2 m from either end of the mat such that the walkway was 8.6 m long in total. This was done to avoid the effects of unexpected acceleration and deceleration. After confirmation of eligibility and anthropometric measurements, the participants performed eight trials of barefoot walking on a walkway continuously at a comfortable pace and then again at a fast pace. A five-minute rest was given between the two conditions. The first 16 steps in each condition were used in the data analysis. If obtrusive or unusual motions were observed, the trial was discarded.

### 2.3. Measurement Variables

The COP excursions were parameterized into three categories as follows: (1) temporal parameters, including four time points and five subphases of the stance phase, (2) spatial parameters including displacement ranges and mean locations, and (3) velocities.

The four time points were midfoot-on time (MON), the time at which the first sensor appears in the middle one-third of the foot; foot flat time (FFT), the time at which 78% of all sensors associated with the foot have been activated; heel-off time (HOT), the time at which the last sensor deactivates in the proximal one-third of the foot; and MOF, the time at which the last sensor deactivates in the middle one-third of the foot. The subphases were divided by time points (Figure 2). All temporal variables were normalized by the duration of the stance phase and were expressed as percentages. Spatial variables were calculated in the mediolateral and anteroposterior directions in each subphase. The mediolateral and anteroposterior mean locations were normalized by the width and length of the subjects’ foot, respectively, and were expressed as percentages. Velocities in the mediolateral and anteroposterior directions and total velocity were calculated in each subphase.

After the statistical analysis, a relatively large amount of variability in FFT was found among the time points, and we analyzed the COP parameters in a combined phase with late contact phase (LCP) and foot flat phase (FFP) as additional variables.

### 2.4. Data Preprocessing

Parallel transference and rotational transform calculations were performed over COP coordinates of all *k* footfalls in the raw time series data with the heel center and toe, which are the pivot points of the proximal and distal one-third of footfall. The original COP coordinate of the *i*th footfall of length *N* is defined as $P_{\mathrm{raw}}^{\left(i\right)}$, wherein i= {1, …, k}. From $P_{\mathrm{raw}}^{\left(i\right)}\left(a_{n},\;b_{n}\right)$, wherein i= {1, …, k}, the sample $a_{n}$ and $b_{n}$ with n ∈{1, …, N} consist of data from the horizontal and the vertical axes of the mat, respectively. The toe center *T* and the heel center *H* of the *i*th footfall are defined as $T^{\left(i\right)}=\left(c,\;d\right)$ and $H^{\left(i\right)}=\left(e,f\right)$ with i= {1, …, k}, respectively, and the angle $\theta_{o}^{\left(i\right)}$ of inclination of a line $l^{\left(i\right)}$ running from $H^{\left(i\right)}$ to $T^{\left(i\right)}$ is $\theta_{o}^{\left(i\right)}=\tan^{-1}\left(\frac{d-f}{c-e}\right)$ with i= {1, …, k}. The angle $\theta_{c}^{\left(i\right)}$ between the vertical axis of the mat and the line can be calculated and converted to coordinate $P_{c}^{\left(i\right)}$ in a new coordinate system having $H^{\left(i\right)}$ as the origin and $l^{\left(i\right)}$ as the *y*-axis is the following matrix (Equation (1)):(1)$$P_{c}^{\left(i\right)}\left(\begin{array}{c} x_{n}\\ y_{n} \end{array}\right)=\left(\begin{array}{cc} \cos\theta_{c}^{\left(i\right)} & -\sin\theta_{c}^{\left(i\right)}\\ \sin\theta_{c}^{\left(i\right)} & \;\cos\theta_{c}^{\left(i\right)} \end{array}\right)\left(\begin{array}{c} a_{n}-e\\ b_{n}-f \end{array}\right).$$

The x- and y-axes of the new coordinate plane were considered as the mediolateral and anteroposterior directions, respectively (Figure 3). All data management was implemented using MATLAB software.

### 2.5. Sample Size

Following the method proposed by Walter et al. [13], the minimum sample size was calculated to be 90 participants to ensure that the expected reliability level of 0.6 exceeded a minimally acceptable level of reliability of 0.5 at a one-tailed α level of 0.05, and a power of 80%.

### 2.6. Statistical Analysis

Descriptive statistics for general characteristics and gait performance were calculated and the Shapiro–Wilk test for normality was performed. Reliability was evaluated using relative and absolute reliability indices. For relative reliability, ICC was used as 2-way random models (ICC_2,1_, ICC_2,*k*_) with absolute agreement [14]. Repeated measures analysis of variance was performed to detect systematic bias [15]. To determine the number of footfalls (*k*) required to obtain the desired reliability (*R_k_*), the following Spearman–Brown formula (Equation (2)) was used [16]:(2)$$R_{k}=\frac{kR}{1+\left(k-1\right)R}$$

From the reliability coefficient (R) estimated from this study, *k* was calculated for *R_k_* using (Equation (3)) the following equation:(3)$$k=\frac{R_{k}\left(1-R\right)}{R\left(1-R_{k}\right)}$$

As a cut-off level of reliability, *R_k_* was set at 0.50 following the criteria classified by Portney LG et al. [17], and the criteria were applied to the interpretation of the ICC for the average measurement (poor to moderate, 0.50–0.75; good, 0.75–0.90; and acceptable, >0.90).

As the absolute reliability index, the coefficient of variance (CV)% was calculated except for the mean COP location parameters, due to inclusion of either positive or negative values in the data set [15]. The standard error of measurement and minimal detectable change were also calculated for the average measurement [18]. Minimal detectable change was calculated by multiplying the standard error of measurement by the square root of 2 and the *z* score of 1.64 at the 90% confidence level [19]. Although the 95% confidence level is commonly used in research requiring precision, the 90% level was judged to be more acceptable in this study because that level is more relevant in clinical decisions and seems to be the most common standard [20]. The square root of 2 represents additional uncertainty in repeated measurements [19]. All statistical analyses were performed only from the data of the dominant foot [21]. However, to confirm the difference in variability between the dominant and non-dominant feet, CV was calculated for both.

## 3. Results

Table 1 shows the general characteristics of the participants and compares the two gait speed parameters between conditions. Significant differences in the parameters were confirmed. Of the participants, 80 (89%) were right-foot dominant, 31 (34%) had a high medial arch of the foot, and 4 (4%) had a low medial arch of the foot.

### 3.1. Relative Reliability and the Number of Footfalls Required

Descriptive statistics, relative reliability, and results of repeated measures analysis of variance for all COP parameters are presented in Table 2. The trajectories and velocities of COP are illustrated in Figure 4 and Figure 5, respectively. Red and green diamond marks indicate the starts of single limb stance phase and double limb stance phase, respectively. Linear interpolation was performed.

A wide range of ICC for a single footfall was found among the parameters, and seven footfalls were determined as the required number to result in more than 90% of the parameters falling under a good to reasonable level of relative reliability. There was no systematic bias in any of the parameters.

### 3.2. Absolute Reliability

The results of CV, standard error of measurement, and minimal detectable change for seven footfalls are presented in Table 3. For all the parameters, the CV values for a comfortable pace and fast pace were 2.7–47.2% and 3.0–59.1%, respectively, in the dominant foot and 2.8–48.3% and 3.1–53.2%, respectively, in the non-dominant foot.

Among the time-point parameters, FFT had the largest variability among all the absolute reliability indices. The duration parameters of LCP and FFP had relatively large variability and those of the combined phase of the two phases had less than half of the variability of the phases.

### 3.3. Foot Dominance and Gait Speed Effects for Variability

The mean differences between the dominant and non-dominant feet in the CV of all parameters were 0.58% and 0.02% at comfortable and fast paces, respectively. Only the duration of FFP at a fast pace showed a relatively large CV difference of 5.9% between the feet.

The mean differences between the two speed conditions in the CV of all the parameters were 1.11% and 0.55% in the dominant and non-dominant feet, respectively. Differences in CV values of FFP duration were 24.1% and 17.1% at comfortable and fast paces, respectively. The differences in the CV values of the anteroposterior displacement range in FFP were 11.1% and 8.5% at comfortable and fast paces, respectively, while that of mediolateral velocity in FFP was 6.6% at a comfortable pace.

There were no other parameters having CV values larger than 5% between the feet or between the conditions.

## 4. Discussion

Segmentation of the stance phase is an important topic for a gait analysis and COP excursion research. In 1999, Han T.R. et al. [22] divided the stance phase into four subphases with an insole system, but it is unclear how the stance phase was divided. They parameterized only one displacement range in the mediolateral and anteroposterior directions and one mean location of the COP. We agree with their argument that the mean location reflects the topographical features of plantar pressure distribution. However, a single coordinate or displacement range cannot be representative of dynamic changes in the progression of the distribution.

In 2005, De Cock et al. [23] defined four subphases of the stance phase with a platform system. The authors reported detailed velocities of COP in each subphase in 2008 [24] and their method was used in studies using a platform. This study provided great insights on the detailed velocities in the mediolateral and anteroposterior directions are informative properties of COP excursion. Nevertheless, the above method is insufficient for observing velocities in the late-stage stance phase during gait. The four subphases were 7.0%, 5.1%, 43.4%, and 44.5% of progression time from the initial to final contact at a 5-km/h walking speed in healthy young adults [25]. A previous study by Cornwall and McPoil [26] presented rapidly increasing velocity of COP within the last subphase, which corresponds to push-off in normal walking. This study reared similar results and reinforced the quick movement by presenting velocities in the mediolateral and anteroposterior directions.

The results presented movement patterns of COP excursion in each subphase during gait. Although there is a limitation that simultaneous measurements such as synchronized motion capture analysis, were not performed, the patterns might be understood through temporal correspondence with related studies. In the initial contact phase (ICP), the COP shifts medially and then laterally with rapidly increasing velocity on the medial side of the foot. The first contact point is normally lateral to the center of the ankle joint, creating a pronation moment at the subtalar joint that allows flexible mobility, for accommodation and shock absorption [27]. The initial pronation and plantar flexion of the calcaneus with respect to the tibia occur rapidly with migration of the load from the opposite foot, and subsequently, the medial arch delivers weight to the lateral border of the foot [28,29,30,31,32].

In LCP, the COP moves laterally with decreasing velocity, with minor increases in the velocity in the mediolateral direction and substantial increases in the anteroposterior direction. This movement can be explained as follows: (a) the movement of the load toward the lateral border of the plantar flexed, inverted, and adducted calcaneus with respect to the tibia to a neutral pose until approximately 28% of the stance phase and (b) reoccurrence of plantar flexion of the calcaneus with inversion and adduction of the cuboid with respect to the calcaneus, and that of the navicular bone with respect to the talus [30,31]. This supination at the transverse tarsal joint provides stability for the joint and along the longitudinal arch [27,31].

The COP moved laterally and then medially, with the least range of mediolateral displacement in the FFP. After FFT, the peak pressure proceeds through the lateral midfoot and the load is distributed to the metatarsal heads with fixation of the forefoot [33]. The leg passes over the foot in midstance, followed by dorsiflexion of the calcaneus with respect to the tibia, and tensing of the plantar aponeurosis known as the windlass mechanism [31,34].

After HOT, with heel rise accompanied by dorsiflexion of the metatarsals and toes, the rearfoot undergoes supination and elongation of the plantar aponeurosis increases quickly to peak at approximately 80% of the stance phase [30,34]. These stabilizing effects of forefoot fixation and windlass mechanism might explain the small amount of variability in both the mediolateral and anteroposterior velocities of the COP in the initial propulsive phase (IPP).

In the late propulsive phase (LPP), the COP continues to shift medially at a high rate. With the start of opposite foot contact, the vertical ground reaction force decreases, the lateral and posterior ground reaction forces increase sharply, and peak pressure occurs in the hallux at midfoot-off time (MOF), which is accompanied by a considerably increasing moment and angle of ankle plantarflexion [32,33,35].

The results of relative reliability showed similar or higher coefficients of ICC to the work of Cornwall and McPoil [26]. However, one parameter of mediolateral displacement range of COP in ICP had lower ICC values. This may be due to underestimation of the coefficients as the ICC is highly affected by the range of the measured values [15]. The parameter had similar absolute reliability indices to other mediolateral displacement range parameters. Although there is no specific cut-off value of CV, the result presented much lower CV values than early reports of conventional methods [24,36]. Therefore, it is relevant that the intra-session reliability of the COP measurement was acceptable.

Early studies reported a negligible asymmetry effect on COP progression [23,25]. Similarly, the present study exhibited few or no asymmetry effects on variability. Walking speed [37] and cadence [38] affect foot pressure distribution. In terms of variability, however, we demonstrated very small differences in CV values between the different speed conditions. Exceptionally, the duration of FFP between the feet, duration, anteroposterior displacement range, and mediolateral velocity in FFP between conditions showed relatively large differences in variability. Because gait speed normally affects the COP velocity in midstance and the midstance duration [39], it is thought to affect the variability, especially of parameters in FFP.

The required number of footfalls was calculated by applying a cut-off score to all COP parameters in both conditions. From the resulting data, more than seven footfalls ensure at least a poor to moderate level of reliability in a healthy young population. The parameters can be selected flexibly, and the number of footfalls to be collected should be determined in accordance with the research methodology.

This work presented the applicability of a mat system for gait analysis on COP excursion. We believe that this method can be useful for populations with impaired dynamic balance, particularly neurological diseases. The mat has a relatively larger active area compared to other gait measurement equipment involved in pressure measurement, which makes it easier for the subject to maintain contact with the measurement equipment. In addition, the mat provides multiple footfalls of both feet in every single measurement trial. It is thought that the simultaneous measurement of the front and rear footfalls in the same foot or of consecutive footfalls in both feet provides great information in the evaluation of dynamic balance during gait.

There are several limitations to this study. First, gait speed was not controlled in any specific range, and the results might not be generalizable in other speed conditions. However, minimal differences in CV between conditions demonstrated the robustness of our results. We intended to capture the individuals’ natural gait patterns and did not limit gait speed. Second, validation of the proposed method was not verified. Although our results showed almost the same results in COP progression as the previous studies and the purpose of the proposed method using a mat system is different from that of conventional methods with other equipment such as force plates, it is necessary to investigate the validity with the conventional equipment. Third, the target population of the study was healthy young adults, and the results cannot be generalizable to other populations. Foot pressure distribution can be affected by many factors such as age, body weight, foot morphology, joint flexibility, and muscular control [1,4] and further studies in other populations are still needed for clinical usage.

## 5. Conclusions

The presented method can be used as a reliable measurement for COP excursion during gait, and it is recommended that more than seven footfalls be collected to ensure a high level of reliability.

## Figures and Tables

**Figure 1 ijerph-17-07845-f001:**
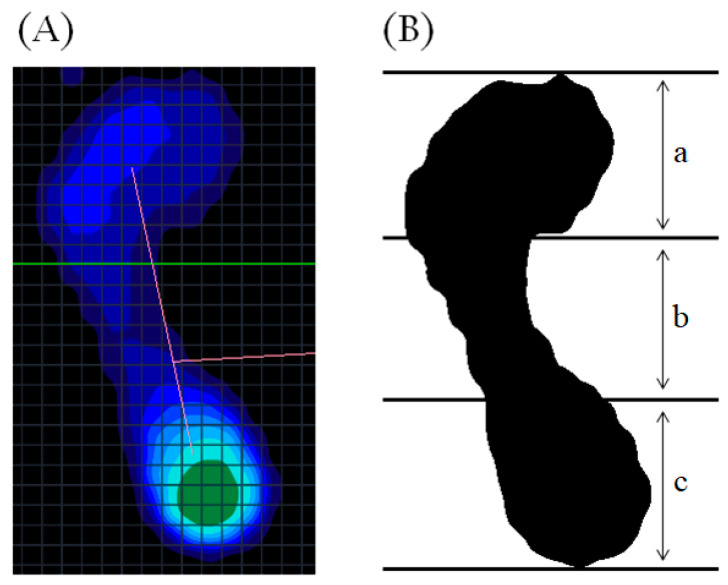
Illustration for calculation of the arch index. (**A**) shows the pressure footprint with exclusion of the toes; and (**B**) is the conversion. Arch index is calculated as the area of middle third of the footprint divided by the entire area (b/(a + b + c)). a, distal one-third of the footprint; b, middle third of the footprint; c, distal one-third of the footprint.

**Figure 2 ijerph-17-07845-f002:**
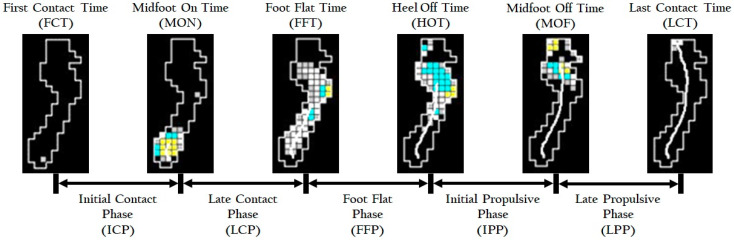
Illustration of temporal variables of the center of pressure measurement.

**Figure 3 ijerph-17-07845-f003:**
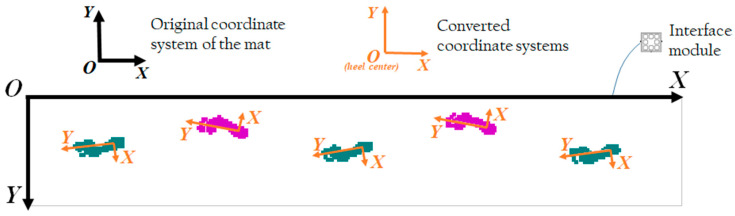
Illustration of the original coordinate plane (black axes) and converted individuals in each footfall (orange axes). Please note that the positive direction of the *x*-axis in the conversion is from the medial to lateral side of the foot.

**Figure 4 ijerph-17-07845-f004:**
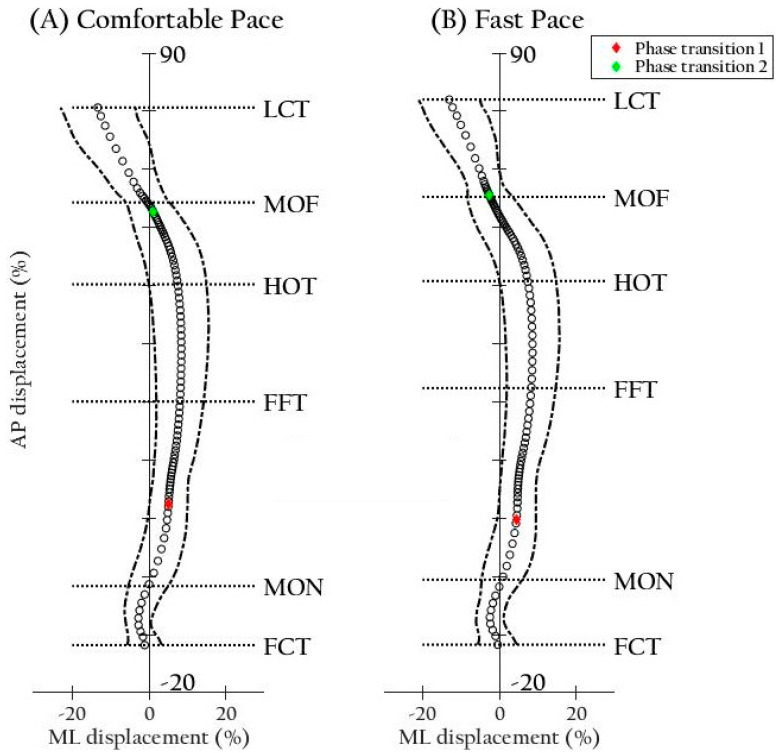
Illustration of center of pressure trajectories (**A**) at a comfortable and (**B**) at a fast pace. One hundred empty circles and the bilateral dashed lines compose a mean trajectory and ± 2 SD on the *x*-axis for all participants. X and Y coordinates were normalized by average width and length, respectively. Abbreviations: AP, anteroposterior; FCT, first contact time; FFT, foot flat time; HOT, heel-off time; LCT, last contact time; ML, mediolateral; MOF, midfoot-off time; MON, midfoot-on time.

**Figure 5 ijerph-17-07845-f005:**
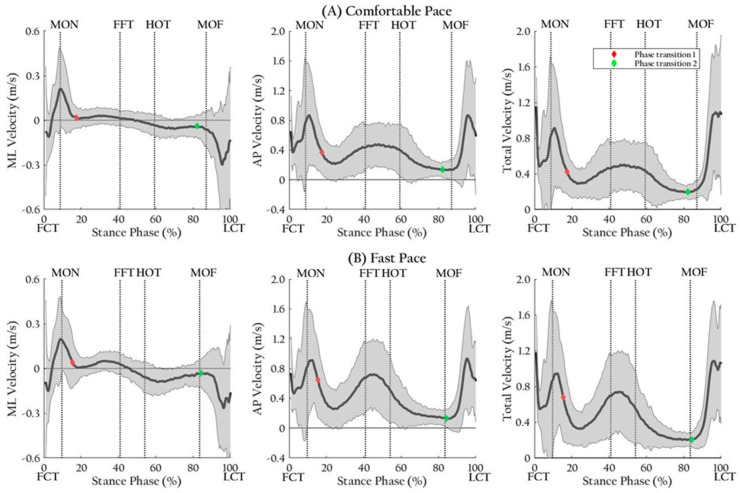
Illustration of velocities of center of pressure excursion. Mediolateral, anteroposterior, and total velocities of center of pressure excursion at a comfortable (**A**) and at a fast pace (**B**) were plotted for all participants. The plots are expressed as mean of a bold line and range within ± 2 SD of the grey area. Abbreviations: AP, anteroposterior; FCT, first contact time; FFT, foot flat time; HOT, heel-off time; LCT, last contact time; ML, mediolateral; MOF, midfoot-off time; MON, midfoot-on time.

**Table 1 ijerph-17-07845-t001:** Anthropometric and general characteristics of the subjects.

Variables		Values	*p* Value ^†^
Sex (Male/Female)		48/42	
Age (years)		22.4 (2.4)	
Height (cm)		167.5 (8.6)	
Body mass (kg)		64.4 (14.3)	
Body mass index (kg/m^2^)		22.9 (3.7)	
Leg Length (cm)		86.5 (4.9)	
Foot length (mm)		240.1 (17.3)	
Foot width (mm)		100.1 (9.9)	
Arch index		0.195 (0.079)	
Walking speed (cm/sec)			
	Comfortable pace	129.0 (16.0)	<0.001 *
	Fast pace	161.8 (20.6)	
Cadence (steps/min)			
	Comfortable pace	118.3 (8.0)	<0.001 *
	Fast pace	134.0 (10.6)	

Values are expressed as mean (SD) or n. ^†^ Significance level for paired *t*-test. * Significant difference between two speed conditions (*p* < 0.001).

**Table 2 ijerph-17-07845-t002:** Summary of descriptive statistics and relative reliability for all COP parameters.

Parameters		Comfortable Pace				Fast Pace			
		Mean (SD)	ICC_(2,1)_ ^†^	ICC_(2,_*_k_*_)_ ^‡^	F Value (*p* Value)	Mean (SD)	ICC_(2,1)_ ^†^	ICC_(2,*k*)_ ^‡^	F Value (*p*-Value)
Duration (%)	ICP	8.83 (2.19)	0.79 (0.74–0.84) *	0.96	0.68 (0.80)	9.71 (2.45)	0.74 (0.68-0.80) *	0.95	1.30 (0.19)
	LCP	32.02 (7.12)	0.46 (0.38–0.54) *	0.86	0.61 (0.87)	31.19 (6.89)	0.44 (0.36–0.52) *	0.84	1.24 (0.23)
	FFP	18.52 (5.73)	0.42 (0.34–0.50) *	0.83	0.55 (0.91)	13.5 (5.84)	0.40 (0.32–0.48) *	0.82	1.27 (0.21)
	CbP ^§^	50.49 (7.01)	0.76 (0.70–0.81) *	0.96	0.77 (0.71)	44.41 (6.27)	0.75 (0.70–0.81) *	0.96	1.24 (0.24)
	IPP	27.69 (5.67)	0.62 (0.55–0.69) *	0.92	0.62 (0.86)	29.38 (4.52)	0.48 (0.40–0.56) *	0.86	1.25 (0.23)
	LPP	12.98 (3.45)	0.55 (0.48–0.63) *	0.90	1.26 (0.22)	16.51 (4.50)	0.55 (0.47–0.63) *	0.89	1.27 (0.21)
Stance duration (ms)		613.7 (46.4)	0.87 (0.84–0.91) *	0.98	1.28 (0.21)	528.6 (46.4)	0.88 (0.85–0.91) *	0.98	0.95 (0.50)
Time-point (%)	MON	8.83 (2.19)	0.79 (0.74–0.84) *	0.96	0.68 (0.80)	9.71 (2.45)	0.74 (0.68–0.80) *	0.95	1.30 (0.19)
	FFT	40.85 (7.33)	0.47 (0.40–0.56) *	0.86	0.59 (0.89)	40.9 (7.30)	0.45 (0.38–0.54) *	0.85	1.05 (0.40)
	HOT	59.32 (7.07)	0.77 (0.71–0.82) *	0.96	0.93 (0.53)	54.11 (6.23)	0.76 (0.70–0.81) *	0.96	1.24 (0.24)
	MOF	87.02 (3.45)	0.55 (0.48–0.63) *	0.90	1.26 (0.22)	83.48 (4.50)	0.55 (0.47–0.63) *	0.89	1.30 (0.19)
Mediolateral Displacement Range (mm)	ICP	4.89 (0.99)	0.13 (0.09–0.18) *	0.50	1.32 (0.18)	4.97 (1.11)	0.17 (0.12–0.23) *	0.58	0.89 (0.57)
	LCP	10.37 (3.50)	0.49 (0.41–0.57) *	0.87	0.85 (0.63)	10.05 (3.27)	0.46 (0.39–0.55) *	0.86	1.29 (0.20)
	FFP	3.57 (1.13)	0.23 (0.17–0.31) *	0.68	1.10 (0.35)	3.31 (1.11)	0.22 (0.16–0.29) *	0.66	1.36 (0.16)
	CbP	12.11 (3.51)	0.46 (0.38–0.55) *	0.86	1.08 (0.37)	11.40 (3.42)	0.48 (0.40–0.56) *	0.86	1.24 (0.23)
	IPP	9.85 (3.50)	0.51 (0.44–0.60) *	0.88	1.17 (0.29)	11.59 (3.45)	0.49 (0.41–0.57) *	0.87	0.88 (0.58)
	LPP	16.16 (3.93)	0.39 (0.32–0.48) *	0.82	1.00 (0.45)	13.99 (3.59)	0.42 (0.35–0.51) *	0.84	0.68 (0.81)
Anteroposterior Displacement Range (mm)	ICP	21.43 (4.02)	0.46 (0.39–0.55) *	0.86	0.46 (0.96)	21.98 (4.14)	0.53 (0.46–0.61) *	0.89	0.82 (0.66)
	LCP	78.04 (14.29)	0.48 (0.41–0.57) *	0.87	0.77 (0.71)	87.09 (18.78)	0.49 (0.41–0.57) *	0.87	1.65 (0.06)
	FFP	55.87 (12.59)	0.41 (0.34–0.50) *	0.83	0.62 (0.86)	49.57 (13.59)	0.34 (0.27–0.43) *	0.78	0.96 (0.49)
	CbP	133.1 (10.7)	0.78 (0.72–0.83) *	0.96	1.51 (0.09)	133.5 (10.9)	0.73 (0.67–0.79) *	0.95	0.98 (0.48)
	IPP	28.86 (8.05)	0.71 (0.65–0.78) *	0.95	0.62 (0.86)	30.44 (7.84)	0.69 (0.62–0.76) *	0.94	0.55 (0.91)
	LPP	39.65 (6.97)	0.60 (0.53–0.68) *	0.91	1.39 (0.14)	40.04 (5.57)	0.59 (0.52–0.67) *	0.91	0.87 (0.60)
Mediolateral Velocity (cm/s)	ICP	20.02 (4.11)	0.22 (0.17–0.30) *	0.67	1.18 (0.28)	20.35 (4.65)	0.25 (0.19–0.33) *	0.70	0.70 (0.79)
	LCP	10.06 (1.81)	0.45 (0.37–0.53) *	0.85	0.89 (0.58)	11.21 (2.59)	0.47 (0.40–0.56) *	0.86	1.50 (0.10)
	FFP	8.05 (1.21)	0.25 (0.19–0.33) *	0.70	0.81 (0.67)	9.83 (1.95)	0.24 (0.18–0.32) *	0.69	1.37 (0.15)
	CbP	9.06 (1.37)	0.52 (0.44–0.60) *	0.88	0.98 (0.48)	10.34 (1.65)	0.51 (0.44–0.60) *	0.88	1.22 (0.25)
	IPP	10.27 (1.12)	0.26 (0.19–0.33) *	0.71	1.52 (0.09)	11.8 (1.76)	0.38 (0.31–0.47) *	0.81	1.15 (0.31)
	LPP	36.45 (9.16)	0.32 (0.26–0.41) *	0.77	1.17 (0.29)	31.63 (7.37)	0.34 (0.27–0.42) *	0.78	1.03 (0.42)
Anteroposterior Velocity (cm/s)	ICP	46.75 (7.79)	0.43 (0.35–0.51) *	0.84	0.84 (0.63)	50.32 (9.57)	0.51 (0.43–0.59) *	0.88	0.73 (0.75)
	LCP	48.61 (10.69)	0.63 (0.56–0.70) *	0.92	0.90 (0.56)	60.00 (11.95)	0.59 (0.51–0.66) *	0.91	1.08 (0.37)
	FFP	54.23 (10.99)	0.50 (0.42–0.58) *	0.87	1.44 (0.12)	74.84 (18.39)	0.55 (0.48–0.63) *	0.90	0.54 (0.92)
	CbP	49.16 (9.09)	0.72 (0.66–0.78) *	0.95	1.50 (0.10)	62.32 (10.76)	0.71 (0.64–0.77) *	0.94	1.66 (0.05)
	IPP	19.82 (4.04)	0.63 (0.56–0.71) *	0.92	0.99 (0.47)	22.18 (4.49)	0.63 (0.56–0.71) *	0.92	1.40 (0.14)
	LPP	55.97 (11.15)	0.40 (0.33–0.48) *	0.82	1.01 (0.44)	52.20 (10.97)	0.45 (0.38–0.54) *	0.85	1.00 (0.45)
Total Velocity (cm/s)	ICP	54.15 (8.85)	0.46 (0.38–0.54) *	0.85	1.27 (0.21)	57.45 (10.84)	0.51 (0.43–0.59) *	0.88	0.75 (0.74)
	LCP	50.45 (10.71)	0.62 (0.55–0.70) *	0.92	0.88 (0.59)	61.87 (12.07)	0.58 (0.51–0.66) *	0.91	1.07 (0.38)
	FFP	55.55 (10.97)	0.50 (0.43–0.59) *	0.88	1.47 (0.11)	76.25 (18.40)	0.55 (0.48–0.64) *	0.90	0.54 (0.92)
	CbP	50.78 (9.09)	0.71 (0.65–0.78) *	0.95	1.46 (0.11)	63.96 (10.73)	0.70 (0.64–0.77) *	0.94	1.65 (0.05)
	IPP	23.78 (3.81)	0.55 (0.48–0.63) *	0.90	1.12 (0.33)	26.78 (4.47)	0.56 (0.49–0.65) *	0.90	1.51 (0.09)
	LPP	72.82 (14.27)	0.36 (0.29–0.45) *	0.80	1.16 (0.30)	66.20 (13.20)	0.41 (0.34–0.50) *	0.83	0.92 (0.54)
Mediolateral Mean Location (%)	ICP	−2.19 (1.40)	0.40 (0.33–0.49) *	0.82	0.95 (0.51)	−1.81 (1.27)	0.36 (0.29–0.45) *	0.80	0.86 (0.61)
	LCP	5.59 (2.00)	0.46 (0.39–0.55) *	0.86	0.91 (0.55)	5.46 (2.10)	0.49 (0.42–0.58) *	0.87	0.80 (0.67)
	FFP	8.09 (2.94)	0.37 (0.30–0.46) *	0.80	1.42 (0.13)	7.75 (2.85)	0.41 (0.34–0.50) *	0.83	0.97 (0.49)
	CbP	6.49 (2.13)	0.43 (0.36–0.52) *	0.84	1.32 (0.18)	6.07 (2.12)	0.50 (0.43–0.59) *	0.88	0.71 (0.77)
	IPP	3.63 (2.80)	0.45 (0.37–0.53) *	0.85	1.29 (0.20)	1.91 (2.52)	0.40 (0.33–0.49) *	0.82	1.47 (0.11)
	LPP	−6.03 (3.05)	0.50 (0.42–0.58) *	0.87	1.20 (0.27)	−6.59 (2.39)	0.41 (0.33–0.49) *	0.83	0.84 (0.63)
Anteroposterior Mean Location (%)	ICP	−7.45 (1.17)	0.40 (0.32–0.48) *	0.82	0.88 (0.59)	−7.12 (1.07)	0.36 (0.29–0.45) *	0.80	1.31 (0.19)
	LCP	16.19 (2.86)	0.47 (0.40–0.56) *	0.86	0.20 (1.0)	17.35 (3.32)	0.45 (0.38–0.54) *	0.85	1.18 (0.28)
	FFP	40.34 (3.71)	0.54 (0.46–0.62) *	0.89	0.58 (0.89)	42.27 (4.24)	0.48 (0.40–0.56) *	0.87	0.80 (0.68)
	CbP	25.22 (2.28)	0.52 (0.44–0.60) *	0.88	0.53 (0.93)	24.63 (2.32)	0.54 (0.46–0.62) *	0.89	1.35 (0.16)
	IPP	59.21 (2.51)	0.72 (0.66–0.79) *	0.95	1.08 (0.37)	60.42 (2.38)	0.70 (0.64–0.77) *	0.94	1.54 (0.08)
	LPP	70.55 (3.35)	0.74 (0.68–0.80) *	0.95	1.62 (0.06)	71.56 (2.93)	0.72 (0.65–0.78) *	0.95	1.32 (0.18)

Abbreviations: FFP, foot flat phase; FFT, foot flat time; HOT, heel-off time; ICP, initial contact phase; IPP, initial propulsive phase; LCP, late contact phase; LPP, late propulsive phase; MOF, midfoot-off time; MON, midfoot-on time. ^†^ Intraclass correlation coefficient for a single footfall. ^‡^ Intraclass correlation coefficient for *k* footfalls when the *k* is seven. ^§^ CbP is the combined phase with LCP and FFP. * Significant at α < 0.001.

**Table 3 ijerph-17-07845-t003:** Absolute reliability for all COP parameters.

Parameters		Comfortable Pace				Fast Pace			
		CV (%)		SEM	MDC	CV (%)		SEM	MDC
		D	ND			D	ND		
Duration (%)	ICP	12.28	12.75	0.46	1.08	14.46	14.46	0.61	1.41
	LCP	22.95	23.39	3.84	8.91	23.83	23.62	3.94	9.14
	FFP	35.02	36.15	3.46	8.02	59.14	53.21	3.73	8.64
	CbP ^†^	7.44	6.80	1.65	3.84	7.48	7.41	1.50	3.48
	IPP	14.88	15.04	2.01	4.65	14.89	15.16	2.33	5.41
	LPP	19.94	20.84	1.46	3.39	21.50	21.03	1.93	4.48
Stance duration (s)		2.72	2.76	7.02	16.27	3.01	3.06	6.71	15.57
Time-point (%)	MON	12.28	12.75	0.46	1.08	14.46	14.46	0.61	1.41
	FFT	17.69	18.10	3.79	8.79	18.52	18.32	4.00	9.29
	HOT	6.25	5.64	1.63	3.77	6.07	5.97	1.48	3.44
	MOF	3.14	3.29	1.46	3.39	4.43	4.31	1.92	4.46
Mediolateral Displacement Range (mm)	ICP	41.46	43.76	1.64	3.80	41.93	42.83	1.53	3.56
	LCP	32.46	32.06	1.76	4.08	32.07	31.87	1.74	4.04
	FFP	47.19	48.29	1.21	2.80	51.85	50.33	1.26	2.91
	CbP	29.71	29.49	1.89	4.39	29.41	29.80	1.76	4.09
	IPP	33.15	30.93	1.64	3.80	28.78	28.12	1.73	4.01
	LPP	29.42	30.94	2.56	5.93	28.99	32.05	2.15	4.98
Anteroposterior Displacement Range (mm)	ICP	18.80	20.13	2.14	4.96	17.03	17.14	1.85	4.28
	LCP	17.43	18.52	7.20	16.70	20.49	20.18	9.42	21.84
	FFP	26.72	27.50	7.74	17.95	37.78	36.01	10.19	23.64
	CbP	3.91	4.20	2.37	5.50	4.13	4.08	2.82	6.53
	IPP	17.62	17.87	2.18	5.05	16.83	18.50	2.27	5.27
	LPP	13.98	14.64	2.60	6.03	11.27	11.94	2.12	4.93
Mediolateral Velocity (cm/s)	ICP	31.62	31.65	4.54	10.52	31.63	30.97	4.64	10.76
	LCP	17.41	18.27	1.01	2.33	18.66	19.07	1.35	3.12
	FFP	21.60	22.79	1.22	2.82	28.18	26.34	2.02	4.69
	CbP	13.12	13.20	0.63	1.47	13.99	13.87	0.77	1.78
	IPP	16.37	17.21	1.11	2.57	16.70	16.40	1.17	2.71
	LPP	31.43	33.20	7.27	16.86	28.82	30.16	5.56	12.89
Anteroposterior Velocity (cm/s)	ICP	16.85	17.63	4.56	10.59	16.38	16.18	4.52	10.48
	LCP	15.08	15.37	3.67	8.51	14.07	15.95	4.61	10.70
	FFP	18.25	19.07	5.32	12.34	19.67	19.96	7.77	18.02
	CbP	10.23	10.41	2.43	5.64	9.70	10.08	2.99	6.94
	IPP	14.73	15.98	1.38	3.19	14.40	15.93	1.53	3.56
	LPP	21.99	23.27	7.10	16.46	21.16	20.95	6.00	13.91
Total Velocity (cm/s)	ICP	15.78	16.76	4.81	11.16	16.07	15.29	5.10	11.83
	LCP	14.73	15.10	3.74	8.68	13.83	15.67	4.70	10.91
	FFP	17.61	18.54	5.26	12.20	19.14	19.51	7.71	17.87
	CbP	9.99	10.18	2.47	5.72	9.43	9.88	3.00	6.96
	IPP	13.40	14.44	1.61	3.74	13.21	14.27	1.83	4.23
	LPP	22.94	24.61	10.03	23.25	21.75	21.67	8.13	18.85
Mediolateral Mean Location (%)	ICP	NA	NA	0.89	2.06	NA	NA	0.89	2.07
	LCP	NA	NA	1.07	2.49	NA	NA	1.04	2.40
	FFP	NA	NA	2.04	4.72	NA	NA	1.74	4.04
	CbP	NA	NA	1.24	2.88	NA	NA	1.02	2.37
	IPP	NA	NA	1.56	3.63	NA	NA	1.59	3.68
	LPP	NA	NA	1.49	3.46	NA	NA	1.49	3.45
Anteroposterior Mean Location (%)	ICP	NA	NA	0.74	1.73	NA	NA	0.76	1.75
	LCP	NA	NA	1.48	3.44	NA	NA	1.83	4.25
	FFP	NA	NA	1.63	3.79	NA	NA	2.17	5.04
	CbP	NA	NA	1.05	2.43	NA	NA	1.02	2.37
	IPP	NA	NA	0.66	1.53	NA	NA	0.66	1.54
	LPP	NA	NA	0.84	1.94	NA	NA	0.79	1.83

Abbreviations: D, dominant side; FFP, foot flat phase; FFT, foot flat time; HOT, heel-off time; ICP, initial contact phase; IPP, initial propulsive phase; LCP, late contact phase; LPP, late propulsive phase; MDC, minimal detectable change; MOF, midfoot-off time; MON, midfoot-on time; NA, not applicable; ND, non-dominant side. SEM, standard error of measurement; ^†^ CbP is the combined phase with LCP and FFP.

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
