# Peer review of "A Novel Method for Gait Analysis on Center of Pressure Excursion Based on a Pressure-Sensitive Mat"

_ijerph, 2020, doi:10.3390/ijerph17217845_

Round 1
Reviewer 1 Report
The aim of this paper is to propose a center of pressure (COP) measurement method using a pressure mat (the Gaitrite mat).
However, the validation of this proposed method is not addressed here. Indeed, the authors should first carry out its validation by comparing the extracted features to reference ones extracted by, e.g., a force plate system considered in the gait analysis field as a gold standard system.
In addition, the authors have to pay attention about the use of adequate papers in the reference list. For example,
> what is the relationship between “the COP excursion reveal balance ability” (L5) and the paper [1] that deals with the “neck pain”? After checking this paper, I found no link between these items,
> what is the relationship between “gait performance” (L6) and the paper [2] that deals with the “neck pain”? After checking this paper, I found no link between these items,
> what is the relationship between “intervention efficacy” (L6) and the paper [3] that deals with the “neck pain”? After checking this paper, I found no link between these items,
> what is the relationship between “foot and ankle function" (L6) and the paper [4] that deals with the “neck pain”? After checking this paper, I found no link between these items,
> There is no link between the paper [16] and the Spearman–Brown formula (L109)!
Author Response
Comments from Reviewer 1:
Comments and Suggestions for Authors
- The aim of this paper is to propose a center of pressure (COP) measurement method using a pressure mat (the Gaitrite mat).
However, the validation of this proposed method is not addressed here. Indeed, the authors should first carry out its validation by comparing the extracted features to reference ones extracted by, e.g., a force plate system considered in the gait analysis field as a gold standard system.
Response:
- We would like to thank you for your good comments and review details. As suggested by the Reviewer, the force plate system is used as a gold standard equipment in the gait analysis field, and is also used as the most standard equipment in the measurement of balance ability. GAITRite, which has high reliability and validity in spatio-temporal variables, 1,2 is currently one of the most commonly used measurement tool in gait analysis worldwide. Although the reports on the validity and reliability of GAITRite is not for the COP measurement, we carried out this study with the expectation that the value created through the use of this tool would be very large. This is the first attempt to quantify the movement of COP in the stance phase using the GAITRite system. We completely agree with your point and will conduct research to verify the validity in further studies.
- In our proposed protocol, at least 4 footfalls are obtained simultaneously with 2 left and right feet with approximately 8m of walking per trial, but the force plate generally obtains 1 independent footfall for each foot. Also, unlike the mat system, measurements on force plates require a strict control; The top of the plate should be covered with a screen to visually block the subject from changing the gait pattern to place the foot on the plate. Within the control, it is difficult to obtain a whole footfall and frequently causes trials to fail.
- Therefore, our measurement method has three unique characteristics; (1) Front and rear footfalls on the same side are related (be measured simultaneously), (2) Left and right footfalls are sequentially related (be measured simultaneously), (3) Easily obtain a large number of footfalls (In this study, at least 16 footfalls in each foot were obtained in a maximum of 8 trials). Direct comparison is difficult because measurement cannot be performed with the force plate in the same setting we propose. However, in the discussion, the results of the previous study were compared, and it was confirmed that the proposed method showed similar results.
1Bilney, B., Morris, M., & Webster, K. (2003). Concurrent related validity of the GAITRite® walkway system for quantification of the spatial and temporal parameters of gait. Gait & posture, 17(1), 68-74.
2McDonough, A. L., Batavia, M., Chen, F. C., Kwon, S., & Ziai, J. (2001). The validity and reliability of the GAITRite system's measurements: A preliminary evaluation. Archives of physical medicine and rehabilitation, 82(3), 419-425.
- In addition, the authors have to pay attention about the use of adequate papers in the reference list. For example,
what is the relationship between “the COP excursion reveal balance ability” (L5) and the paper [1] that deals with the “neck pain”? After checking this paper, I found no link between these items,
what is the relationship between “gait performance” (L6) and the paper [2] that deals with the “neck pain”? After checking this paper, I found no link between these items,
what is the relationship between “intervention efficacy” (L6) and the paper [3] that deals with the “neck pain”? After checking this paper, I found no link between these items,
what is the relationship between “foot and ankle function" (L6) and the paper [4] that deals with the “neck pain”? After checking this paper, I found no link between these items,
There is no link between the paper [16] and the Spearman–Brown formula (L109)!
Response:
- We apologize for making this mistake. As you pointed out, we made a mistake in matching references related to the content, and you pointed it out well. We checked that the contents of all the literature and the suggested sentences are consistent throughout the text, and we have corrected them all. We hope that you will be satisfied with our revised contents.
Reviewer 2 Report
In this manuscript, the authors propose a new method to quantify center-of-pressure excursion using GaitRite mat, a popular ambulatory gait measurement system. This is well-written manuscript. I only have a few minor comments which I hope the authors consider to improve their manuscript.
Introduction
Line 9: Please clarify what the authors meant by "it cannot measure true vertical force".
Line 18: Similar to the previous comment, please clarify what the authors meant by "with the less data distortion".
Throughout Methods and Results
In general, methods and results well-present. But there are too many abbreviations. Please minimize the use of abbreviations.
Table 1: Line shape is odd. Please fix.
Discussion
Please consider to add a paragraph regarding the implication of this study. What should be the target population that the authors believe to be beneficial from their findings. And why do the authors believe so.
Author Response
Comments from Reviewer 2:
Introduction
- Line 9: Please clarify what the authors meant by "it cannot measure true vertical force".
Response:
- Thank you for your careful point. The sentence was corrected to be understood intuitively as follows:
- In line 9: “it measures distorted vertical force”
- Line 18: Similar to the previous comment, please clarify what the authors meant by "with the less data distortion".
Response:
- As pointed out, the transmission of the meaning was not only ambiguous, but it was an unnecessary sentence, so the content was deleted.
Throughout Methods and Results
- In general, methods and results well-present. But there are too many abbreviations. Please minimize the use of abbreviations.
Response:
- Based on your good points, we tried to minimize the use of abbreviations throughout the entire manuscript. I think the readability of our manuscript has been improved after the correction. Once again thank you for the good suggestion.
- Table 1: Line shape is odd. Please fix.
Response:
- All authors appreciate the Reviewer’s points. As per your suggestion, we fix the line of Table 1:
Discussion
- Please consider to add a paragraph regarding the implication of this study. What should be the target population that the authors believe to be beneficial from their findings. And why do the authors believe so.
Response:
- The Reviewer thought we had the target population in mind when designing the study, and he pointed this out well. We acknowledge that this has not been expressed in the previous manuscript, and once again thank you for your comments for allowing us to further explain this content. We think this revision further improved the quality of our research. All authors endeavored to clearly express our intentions and values for this analysis method, and we look forward to more efficient and active use of our proposed method in research and fields related to the gait analysis.
- In line 259-266: “This work presented the applicability of a mat system for gait analysis on COP excursion. We believe that this method can be useful for populations with impaired dynamic balance, particularly neurological diseases. The mat has a relatively larger active area compared to other gait measurement equipment involved in pressure measurement, which makes it easier for the subject to maintain contact with the measurement equipment. In addition, the mat provides multiple footfalls of both feet in every single measurement trial. It is thought that the simultaneous measurement of the front and rear footfalls in the same foot or of consecutive footfalls in both feet provides great information in the evaluation of dynamic balance during gait.”
Reviewer 3 Report
The aim of the reviewed article to propose a COP measurement method based on a pressure-sensitive mat enabling examination of more detailed characteristics of 26 the excursion throughout the stance phase. The second goal of the study was to determine the minimum number of steps to obtain reliable measurements.
The methodology has been correctly described in the article the studied group is representative. The results were well presented, correctly conducted statistical analyzes - well-chosen tests. The sample size was estimated prior to testing - based on an acceptable reliability level of 0.5 and the power of the test of at least 0.8.
I refer the article to corrections, taking into account which the article will probably be accepted for publication.
My comments:
- Please indicate clearly the work limitations in the Discussion chapter.
- Expand the shortcuts below figure 5
- What is the measurement error of Arch Index calculation based on the used measuring mats? What are the dimensions of a single pressure sensor placed in the pressure-sensitive mat?
- In the article, there are 32 references in the bibliography, while the content of the article contains references to 37 items - there are errors in references, e.g. under Cornwall and McPoil [26] there is no reference to [26] of the article by this author - please, correction or supplementation the bibliography.
Author Response
Comments from Reviewer 3:
Introduction
- Please indicate clearly the work limitations in the Discussion chapter.
Response:
- Thank you for your good point. We confirmed that the description was ambiguous and reorganize the limitations as follows:
- In line 267-274: “There are several limitations to this study. First, gait speed was not controlled in any specific range and the results might not be generalizable in other speed conditions. However, minimal differences in CV between conditions demonstrated the robustness of our results. We intended to capture the individuals’ natural gait patterns and did not limit gait speed. Second, the target population of the study was healthy young adults, and the results cannot be generalizable to other populations. Foot pressure distribution can be affected by many factors such as age, body weight, foot morphology, joint flexibility, and muscular control [1, 4] and further studies in other populations are still needed for clinical usage.”
- Expand the shortcuts below figure 5
Response:
- Unfortunately, we did not understand your point very well. We modified the representation of the legend for the Figure 5, and also slightly increased the gap between the Fig and the text. We hope that our modifications are satisfactory to you.
- What is the measurement error of Arch Index calculation based on the used measuring mats? What are the dimensions of a single pressure sensor placed in the pressure-sensitive mat?
Response:
- The arch index was not obtained using a measuring mat, but was obtained using a foot plate (FDM-S; zebris Medical GmbH, Germany) according to the method presented in the previous study. Since it was obtained by the general characteristics of the subject, not the main outcome variable, we did not try to find the error in the measurement. The dimension of a single pressure sensor placed in the pressure-sensitive mat could not be confirmed in the manual, but the spatial resolution (1.27 cm) was confirmed and presented in the text.
- In the article, there are 32 references in the bibliography, while the content of the article contains references to 37 items - there are errors in references, e.g. under Cornwall and McPoil [26] there is no reference to [26] of the article by this author - please, correction or supplementation the bibliography.
Response:
- We apologize for making this mistake. As you pointed out, we made a mistake in matching references related to the content, and you pointed it out well. We checked that the contents of all the literature and the suggested sentences are consistent throughout the text, and we have corrected them all. We hope that you will be satisfied with our revised contents.
Round 2
Reviewer 1 Report
I would like to thank the authors for their answers.
The authors revised the references substantially in line with the reviewer comments.
Concerning the need of validating the proposed method, I agree with the authors about the difficulty of using a single plat force system for that purpose. Nevertheless, while I am convinced about the benefit of this proposed method, it is of paramount importance to conduct its validation. For this, the authors could adapt a specific protocol in a future study that can let them comparing against a force plate system. In the present paper, I recommend that some words about this matter should be added in the text (ex., in the discussion section, as a limitation to be handled in a future study).
Author Response
Response:
- We would like to thank you for your good comments. The authors agree with your point and added content for the need of validation in the limitations as follows.
In line 267-271: “Second, validation of the proposed method was not verified. Although our results showed almost the same results in COP progression as the previous studies and the purpose of the proposed method using a mat system is different from that of conventional methods with other equipment such as force plates, it is necessary to investigate the validity with the conventional equipment.”
